# An Efficient Text Detection Model for Street Signs

Manhuai Lu [1],* , Yuanxiang Mou [2] , Chin-Ling Chen [3,4,5],* and Qiting Tang [6],*

1 College of Mechanical and Electrical Engineering, University of Electronic Science and Technology of China, Zhongshan Institute, Zhongshan 528400, China
2 School of Mechanical and Electrical Engineering, University of Electronic Science and Technology of China, Chengdu 610054, China; 1966764053@std.uestc.edu.cn
3 College of Computer and Information Engineering, Xiamen University of Technology, Xiamen 361024, China
4 School of Information Engineering, Changchun Sci-Tech University, Changchun 130600, China
5 Department of Computer Science and Information Engineering, Chaoyang University of Technology, Taichung 413310, Taiwan
6 School of Electron and Information Engineering, University of Electronic Science and Technology of China, Zhongshan Institute, Zhongshan 528400, China
* Correspondence: lumanhuai@gmail.com (M.L.); clc@mail.cyut.edu.tw (C.-L.C.); manhuailu@gmail.com (Q.T.)

**Abstract:** Text detection in natural scenes is a current research hotspot. The Efficient and Accurate Scene Text (EAST) detector model has fast detection speed and good performance but is ineffective in detecting long text regions owing to its small receptive field. In this study, we built upon the EAST model by improving the bounding box's shrinking algorithm to make the model more accurate in predicting short edges of text regions; altering the loss function from balanced cross-entropy to Focal loss; improving the model's learning ability on hard, positive examples; and adding a feature enhancement module (FEM) to increase the receptive field of the EAST model and enhance its detection ability for long text regions. The improved EAST model achieved better detection results on both the ICDAR2015 dataset and the Street Sign Text Detection (SSTD) dataset proposed in this paper. The precision and F1 scores of the model also demonstrated advantages over other models on the ICDAR2015 dataset. A comparison of the text detection effects between the improved EAST model and the EAST model showed that the proposed FEM was more effective in increasing the EAST detector's receptive field, which indicates that it can improve the detection of long text regions.

**Keywords:** natural scenes; EAST model; street sign text detection; receptive field; sample balance

## 1. Introduction

At present, advancements in computer vision technology, like license plate recognition, have made everyday life significantly more convenient. With the popularity of mobile devices, such as smartphones, anyone can now easily obtain images and videos of natural scenes using cell phones or digital cameras and share them on the Internet. Among the many objects contained in these images and videos, textual information has an important role. Text information contains rich, precise, and high-level information that gives meaning to the objects in natural scenes, helping people better access and understand the information within images and videos [1]. Therefore, acquiring textual content from images of natural scenes has become a necessary task for machine learning.

A large amount of text in natural scenes contains a wealth of information that can be used to improve people's productivity and facilitate daily life. For example, text on street signs contains geographic location information, which can be used to locate and navigate vehicles; text on store signs contains information on the store's services, which helps customers quickly decide which store to enter; text on product packaging contains detailed information about the products, helping consumers determine, for instance, whether the products are expired. In the face of massive quantities of image data, how to efficiently obtain text from images has become an active area of recent research.

Text detection and recognition technologies for the natural scene of the text in street signs have several applications: (1) for autonomous vehicles, using text information from street signs can assist in vehicle positioning and navigation; (2) for intelligent transportation systems, text information from street signs is an important part of the intelligent transportation system, it can provide geographic location information for the establishment of the intelligent transportation systems; (3) for automated data entry, when constructing an intelligent transportation system's database, intelligent detection and recognition of street sign text can automate traffic data entry, improving productivity and replacing labor-intensive manual entry; (4) for intelligent translation and text-to-speech technology, detecting and recognizing street sign text can greatly assist in navigation for the sight-impaired or people who do not speak a local language, thus facilitating daily life. As the number of vehicles continues to increase, traffic congestion has become a major problem for many cities, and intelligent transportation systems and autonomous driving technologies present themselves as effective solutions. Because of the importance of street sign text detection and recognition for intelligent transportation systems, autonomous driving technology, and helping determine the general utility of text recognition, a study was conducted to make a feasible proposal for text detection in street sign scenes.

The sections of this paper are structured as follows: Section 1 introduces the background and significance of research on text detection in natural scenes and street scenes; Section 2 analyzes the challenges and existing results of text detection in natural scenes and street sign scenes; Section 3 proposes corresponding points for improvement based on the shortcomings of the EAST model; Section 4 analyzes and verifies the modified EAST model through experimental results and comparison with the original EAST model on the ICDAR2015 dataset [2] and the SSTD dataset proposed in this paper; and finally, Section 5 summarizes this paper's findings and proposes directions for future research.

## 2. Related Work

For detecting text regions in natural scenes, existing methods can be broadly classified into two categories: methods based on text features and methods based on deep learning. Methods based on text features can be further divided into those based on connected component analysis and those based on sliding windows. Methods based on connected component analysis tend to detect text regions by designing feature extraction algorithms, among which the stroke width transformation (SWT) [3] and maximum stable extremal regions (MSER) [4] algorithms are representative. Methods based on sliding windows [5–7] slide a window over the original image, use a classifier to determine whether the window contains text, and finally post-process using non-maximum suppression. For example, Lee et al. [5] use six different categories of text features to build weak classifiers, following which they construct a strong classifier from a combination of weak classifiers.

Although detection methods based on text features have achieved a certain amount of success, their results are not satisfactory when facing complex natural scenes. The background of natural scenes is complex, containing vehicles, buildings, trees, pedestrians, and other noises that can interfere with detection. The fonts, sizes, and colors of text in natural scenes also vary. Finally, images of natural scenes are affected by the shooting environment, and there may be problems with different resolutions and blurred images caused by noise. In the face of these challenges, deep-learning-based methods have become the mainstream method and area of research for natural scene text detection. The models designed by deep-learning-based methods can be further divided into two categories: region proposal methods and semantic segmentation methods. Region proposal methods are usually built on classical target detection algorithms, such as Region-Based Convolutional Neural Networks (R-CNN) [8], Faster R-CNN [9], Single Shot MultiBox Detector (SSD) [10], and You Only Look Once (YOLO) [11].

Liao et al. [12] proposed a network structure of TextBoxes based on the SSD model to detect horizontal text regions by setting six candidate bounding boxes with different aspect ratios. Later, Liao et al. [13] proposed the TextBoxes++ model based on TextBoxes,

which can detect text regions in any direction by adding angle information to the predicted bounding boxes. Tian et al. [14] proposed the Connectionist Text Proposal Network (CTPN) model based on Faster R-CNN, which treats text regions as sequential information and introduces CNNs for processing. CTPN is better for long text region detection, but the introduction of CNNs greatly increases the number of parameters of the model, and the model is not effective in detecting text regions with high tilt angles. Shi et al. [15] proposed the SegLink model based on CTPN, which divides text region detection into two parts: detection and linking. SegLink can link eight directions during linking, which resolves CTPN's issues with detecting high-tilt-angle text regions, but is also susceptible to detecting separate pieces as a whole for denser text regions. Semantic segmentation approaches [16–19] often draw on classical semantic segmentation algorithms, such as Fully Convolutional Networks (FCN) [20] and Feature Pyramid Networks for Object Detection (FPN) [21], which use deep neural networks to extract multi-layer feature maps, perform multi-level feature fusion to obtain a feature map containing rich information, and then predict target segmentation results based on the fused feature maps. The TextField model, proposed by Xu et al. [19], uses directional fields to represent the features of text regions to detect irregularly shaped text. Hu et al. [22] argued that character-level detection is needed in scenarios such as mathematical formulas, but current datasets are text line-level or word-level annotations. To correct this issue, they proposed a WordSup model that can be trained with weak supervision to perform character-level detection on text-line and word-level annotated datasets. Cao et al. [23] Ma et al. [24] introduced bidirectional feature pyramids to enhance the model's learning ability and achieved better detection results, but bidirectional feature pyramids significantly increase the computational load of model training and are difficult to use in practice. Ma et al. [24] introduced graph CNNs to improve the prediction of link relationships between text primitives and the detection of arbitrarily shaped text regions. However, the introduction of graph CNNs brought with it greater parameters and computational effort. Nagaoka et al. [25] proposed a novel text detection CNN architecture sensitive to text scale to improve the prediction of small texts. However, multiple RPN modules complicated the module structure and introduced more parameters, and the module needed to be pre-designed with anchors. Table 1 summarizes the strengths and weaknesses of previously studied text region detection models.

**Table 1.** Advantages and disadvantages of select text region detection models.

| Item | Year | Model | Advantages | Disadvantages |
|:---:|:---:|:---:|:---:|:---:|
| 1 | 2016 | CTPN [14] | Good detection for long text areas | Cannot detect text areas with large tilt angles; a large number of parameters and operations |
| 2 | 2017 | SegLink [15] | Can detect inclined text areas | Easily misidentifies dense text as a single whole area |
| 3 | 2018 | TextField [21] | Can detect text areas with curvature | The post-processing process is complicated |
| 4 | 2021 | Cao et al. [23] | Good detection of quadrilateral text areas | Bidirectional feature pyramid leads to a large number of model parameters and computations |
| 5 | 2021 | Ma et al. [24] | Good detection of arbitrarily shaped text areas | The introduction of graph convolutional neural networks leads to a high parameter number and computation |
| 6 | 2021 | Nagaoka et al. [25] | Good detect small texts | Multiple RPN modules complicate the module structure and introduce more parameters; need pre-designed anchors |

Analysis of classical models and literature [23,24] suggests that current text region detection models are steadily improving, and the shapes of text regions that can be detected are becoming increasingly broad. However, these enhancements come at the cost of increasingly complex network structures, larger models, a larger number of parameters,

and more computational effort, making them difficult to apply in practice. Therefore, the goal of this study was to propose a text region detection model with the simplest possible structure, smaller parameter numbers, less computation, and better performance for the detection and recognition of text in street sign scenes.

At present, research on text detection in natural scenes has made progress, but research on text detection in street signs is still lacking. In addition to the problems that exist with general natural scene text recognition, the following problems exist for street sign scene text detection.

(1)   Street signs contain multiple types of text, such as Chinese characters, English, numbers, and punctuation marks.
(2)   Differences in image brightness can be significant owing to backlit shooting or nighttime shooting.
(3)   Because of different shooting angles, the tilt angle on the street sign text area may be large, leading to a perspective phenomenon.

Classic text region detection models [11,12] tend to divide text region detection into multiple stages, separately detecting sub-sections of the whole region before merging, which is insufficiently accurate and highly time-consuming. In contrast, the EAST model proposed by Zhou et al. [18] simplifies the intermediate process steps and achieves end-to-end text region detection, which greatly improves text detection accuracy and speed, and can detect quadrilateral regions with a large tilt angle in street signs. However, the EAST model has a small receptive field and is not effective in detecting long text regions.

Therefore, this paper presents an improved EAST model, modified from the original EAST model in the following aspects: an improved shrinking algorithm for the bounding box while analyzing text region shape features, an added feature enhancement module (FEM) to increase the model's receptive field and ability to extract features for long text regions, and a Focal loss function to address imbalances between numbers of positive and negative samples and hard versus easy samples for text region detection. The effect of the improved EAST model is then discussed on the public ICDAR2015 and the SSTD datasets.

## 3. Improving the East Model

The overall flow of the EAST model is shown in Figure 1. First, EAST generates the quadrangles of the dataset, expressed by the smallest external rectangle of the bounding box and the angle between that rectangle and the horizontal plane. The area used to extract features is then shrunken by a certain ratio. The images are fed into a Fully Convolutional Network (FCN) for feature extraction and merging, and then the results corresponding to the label geometry (rotated box or quadrangle) are output. Finally, the gap between labeled information and the output result, i.e., the loss function, is calculated. The loss function calculation result is used to update the parameters of the network. In this paper, we establish that the shortcomings of the EAST model are threefold: in the label generation process, the shrinkage ratio of the bounding box quadrangle is unreasonably set to 0.3 for both long and short edges; the receptive field for the network's feature extraction is insufficient, making it difficult to extract relevant feature information between distant parts of a longer text region; the loss function is set to the balanced cross-entropy loss function, which faces issues with resolving imbalances of sample difficulty in the dataset. In the following sections, we propose improvements to address these three shortcomings.

### 3.1. Label Generation

As shown in Figure 2, the labeled region of the EAST model is first shrunk by 0.3 on the long side and then by 0.3 on the short side, and the resulting region is used for extracting features, which can reduce interference factors introduced by labeling errors. One feature to be learned from this region is the location of the smallest outer rectangle of the bounding box, which is measured by the distance from each pixel location to the four edges. Another feature to be learned from this region is the rotation angle between this minimized outer rectangle and the horizontal plane.

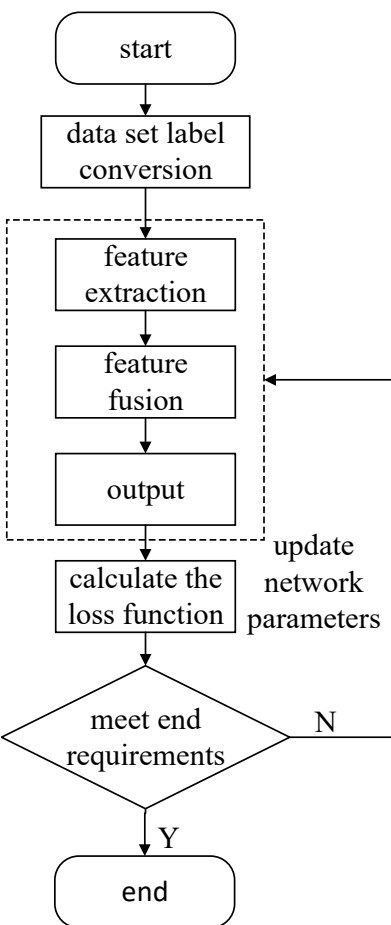

**Figure 1.** Overall flow of the EAST model.

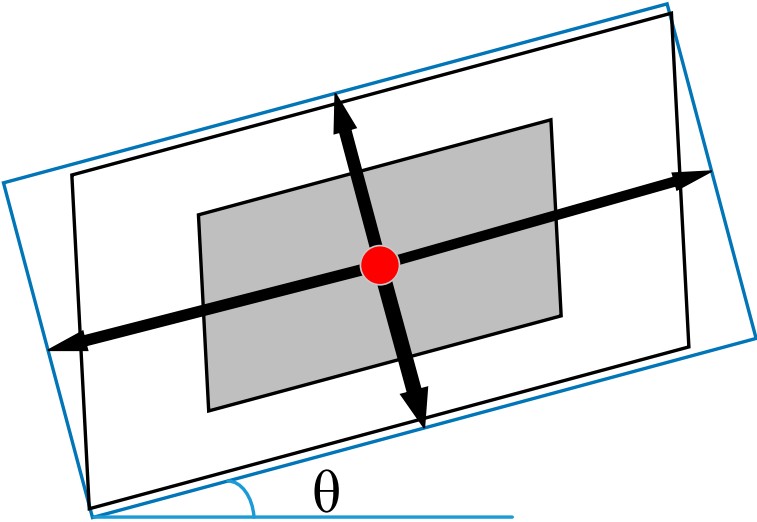

**Figure 2.** Label geometry of EAST.

The EAST model has moderate success by shrinking the long and short edges by 0.3 to reduce labeling error; however, the following problems remain. For a small text area, shrinking the short edge by 0.3 will lead to a large change in the features of Chinese characters and some English letters in upper- and lower-case formats, which affects feature extraction from the text area and eventually leads to an inaccurate prediction bounding box for the text area. As shown in Figure 3, for letters such as *p*, g, and i, shrinking the short edge by 0.3 changes the meaning of letters in the region, leading the model to output inaccurate

prediction boxes. Therefore, this paper proposes improving the shrinkage algorithm of the bounding box by retaining the shrinkage ratio of 0.3 for long edges and reducing the shrinkage ratio to 0.1 for short edges. The final shrinkage strategy algorithm is as follows: input the vertices coordinates of the text region; determine which pair of edges is the long edge and which pair is the short edge by calculating the Euclidean distance between the vertices of adjacent corners; shrink the long edge by 0.3 and the short edge by 0.1; and finally, output the new region's shrunken coordinates of the vertices.

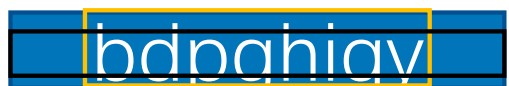

**Figure 3.** Example of problems with the original bounding box shrinking algorithm.

### 3.2. Improving the Network Structure of the EAST Model

The network structure of the EAST model can be roughly divided into three parts [18]: feature extractor stem, feature-merging branch, and output layer. The EAST model's network structure is shown in Figure 4a. The EAST detector uses PVANet as its backbone network to extract features and obtains four levels of feature maps, f1, f2, f3, and f4, with sizes of 1/4, 1/8, 1/16, and 1/32 of the input image, respectively. Larger feature maps have shallower depths and smaller receptive fields, and contain more details, and are used to extract features of small targets; smaller feature maps have larger depths and receptive fields, contain more high-level semantic information, and are used to extract features of large targets. The feature maps f1, f2, f3, and f4 are then merged using the U-shape idea to obtain a feature map that contains both high-level and low-level information. This feature map is then fed into the output layer, and, by mapping across different convolutional layers, the network obtains a score map and a multi-channel geometry map. The geometry feature map contains two parts: RBOX geometry, with axis-aligned bounding box (AABB) information and rotation angle information, and QUAD geometry, with eight channels of coordinate shift information.

The classic VGG16 network proposed by Simonyan et al. [26] is used for the feature extraction stem of the network, removing the final fully connected layer and only preserving the front portion of the full convolutional layer to constitute the core network. However, the theoretical receptive field of the VGG16 network is only 212, and the actual receptive field is smaller, hampering feature extraction for large-scale text regions with input size of $512 \times 512$ pixels. Therefore, this paper proposes an FEM to increase the receptive field of the model. The network structure of the improved EAST model is shown in Figure 4b.

The FEM is inspired by dilated convolution, proposed by Yu et al. [27], and the Inception structure, proposed by Szegedy et al. [28,29]. A brief description is given below.

As shown in Figure 5a is a $3 \times 3$ convolution with a dilation factor of 1, with a receptive field per element of $3 \times 3$, and Figure 5b is a $3 \times 3$ convolution layer with a dilation factor of 2, with a receptive field per element of $7 \times 7$. Dilated convolution can increase the receptive field exponentially without exponentially increasing the number of parameters.

Szegedy et al. [28] proposed a structure that aggregates multiple features at different scales, merges them, and then outputs them, namely the Inception V1 module, which can widen the network and control parameter numbers when extracting high-dimensional features. In 2016, Szegedy et al. [29] improved on the previous model by proposing that a $5 \times 5$ convolution can be decomposed into two $3 \times 3$ convolutions, and an $n \times n$ convolution can be decomposed into a combination of a $1 \times n$ convolution and an $n \times 1$ convolution to further reduce parameters, such that a convolution kernel's number of parameters can be reduced from $n \times n$ to $2n$.

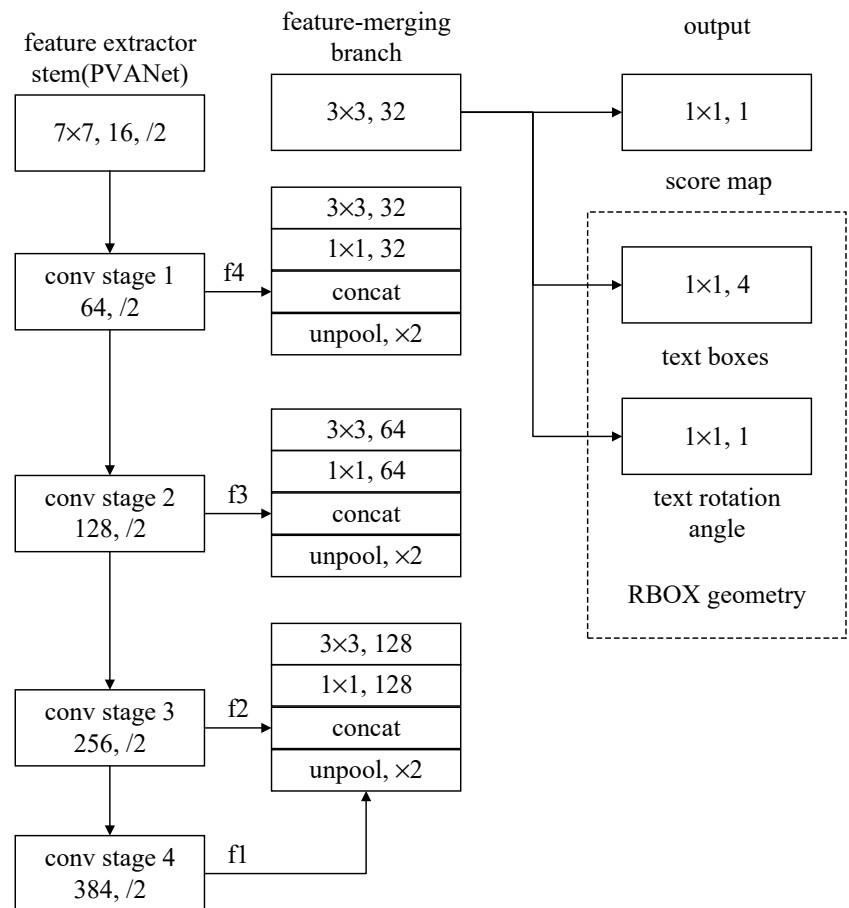

(**a**) The EAST model's network structure

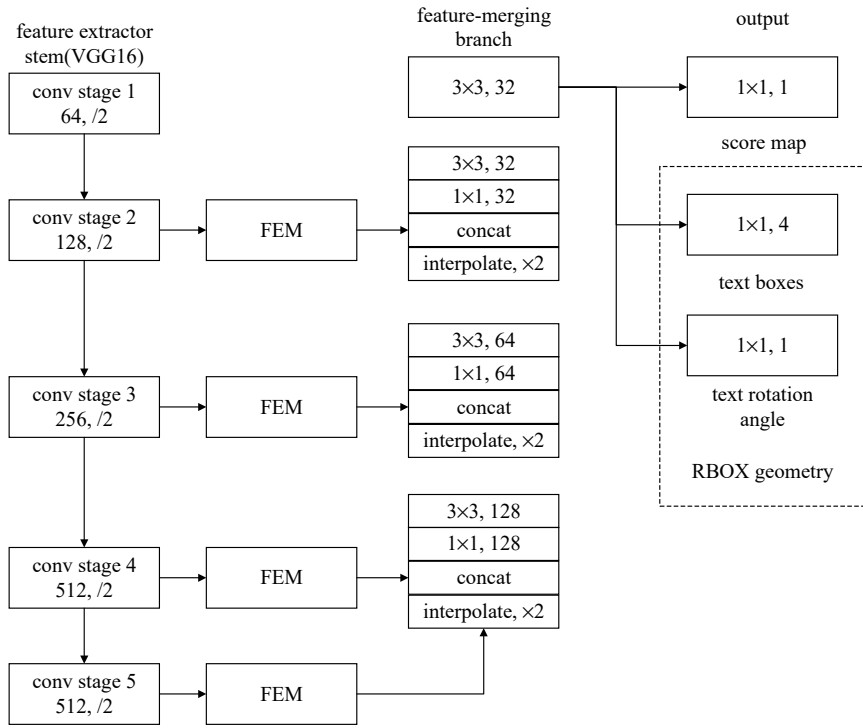

(**b**) The network structure of the improved EAST model

**Figure 4.** The network structure of the EAST model and improved EAST model.

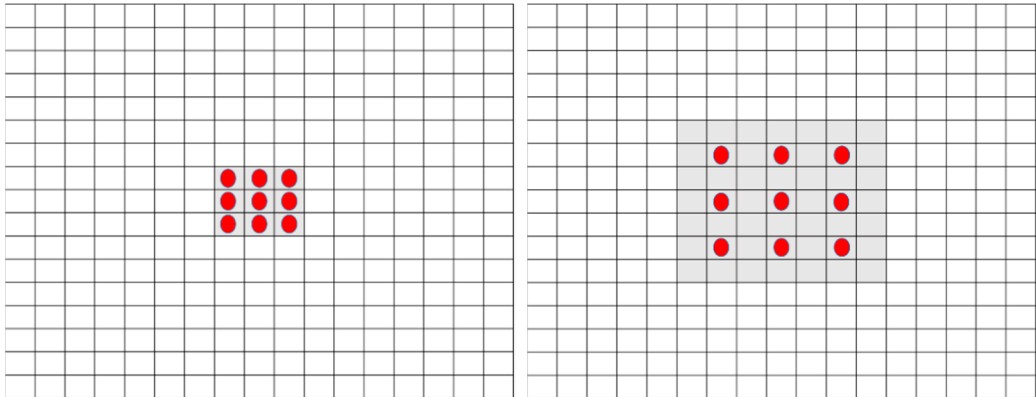

(**a**) The receptive field of 3 × 3 convolution. with a dilation factor of 2.

(**b**) The receptive field of 3 × 3 convolution

**Figure 5.** Receptive field of dilated convolution.

The structure of the FEM designed in this study is depicted in Figure 6. Inspired by the Inception module, the FEM extracts features using a total of four sizes of convolution kernels, $1 \times 1$, $3 \times 3$, $5 \times 5$, and $7 \times 7$, and then merges the extracted features. Considering the reduced number of parameters, the convolution kernels of sizes $3 \times 3$, $5 \times 5$, and $7 \times 7$ are split; that is, the convolution kernel of $3 \times 3$ can be split into a combination of $1 \times 3$ and $3 \times 1$. Moreover, long strip-like convolution kernels like $1 \times 3$ are also more similar to the shapes of text regions in daily life, which is beneficial for feature extraction from text regions. In the structure of Figure 6, except for the $1 \times 1$ convolution kernel, all the other convolution kernels set the dilation factor to 2. Since the FEM in this study mainly contains a few convolutional layers with a small number of parameters, the number of additional parameters does not add too much computational burden.

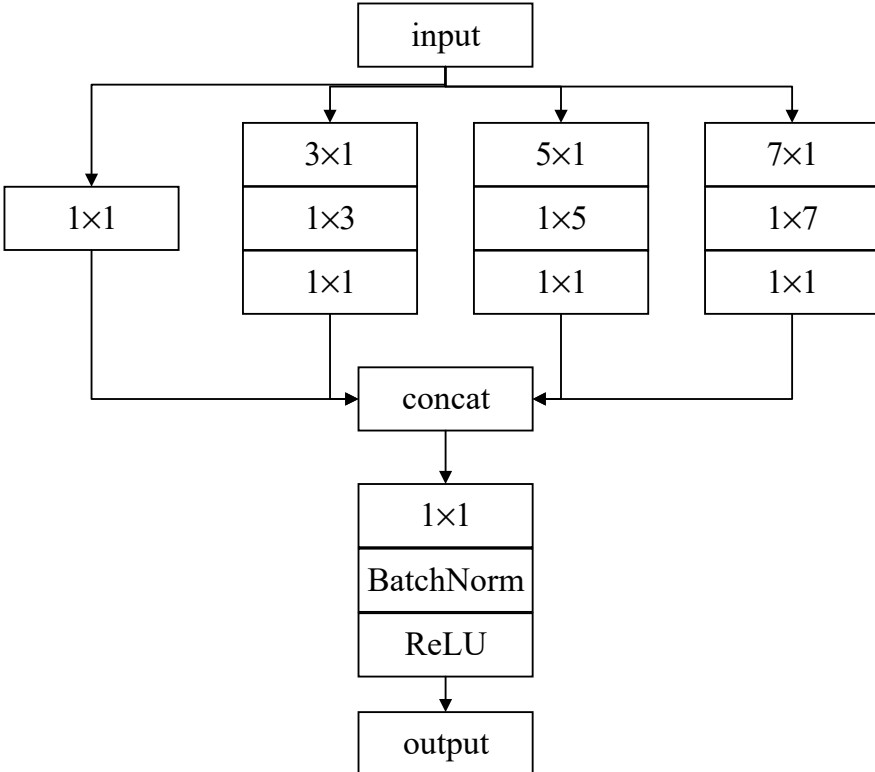

**Figure 6.** Structure of feature enhancement module.

### 3.3. Loss Function

The loss function has two parts [18]: the score map loss function, $L_s$, and the geometries' loss function, $L_g$. The formula is as follows:

$$L = L_s + \lambda_g L_g \tag{1}$$

where $\lambda_g$ represents the importance between two losses, which is set to 1 by the EAST model.

The geometric loss function consists of two parts: the AABB loss function, $L_{AABB}$, and the angle loss function, $L_\theta$. The AABB loss function is as follows:

$$L_{AABB} = -\log IoU(\hat{R}, R^*) = -\log \frac{\left| \hat{R} \cap R^* \right|}{\left| \hat{R} \cup R^* \right|} \tag{2}$$

where $\hat{R}$ represents the predicted AABB geometry and $R^*$ represents the true geometry. The loss function of the rotation angle is calculated as follows:

$$L_\theta(\hat{\theta}, \theta^*) = 1 - \cos(\hat{\theta} - \theta^*) \tag{3}$$

where $\hat{\theta}$ represents the predicted rotation angle and $\theta^*$ represents the true rotation angle. The total geometric loss function is calculated as follows:

$$L_g = L_{AABB} + \lambda_\theta L_\theta \tag{4}$$

where $\lambda_\theta$ is the balance factor, which is set to 10 according to the literature [18].

One of the problems in text region detection is the imbalance in the number of positive and negative samples and the number of hard and easy samples. In the images of the ICDAR2015 dataset, the text region area accounts for less than 5% of the image area. A large amount of background and few regions containing text create an imbalance of positive and negative samples. As shown in Figure 7, the predicted bounding boxes can be divided into positive and negative samples usually. Boxes with an intersection-over-union (IoU) ratio between predicted and true boxes greater than the threshold (usually set to 0.5) are considered positive samples, and boxes with an IoU less than the threshold are considered negative samples. Most of the predicted bounding boxes are not found where the real boxes overlap with the background. These are considered simple samples, which can then be classified as simple positive samples and simple negative samples. As the number of simple negative samples constitutes the largest proportion of all samples, the loss function is dominated by simple negative samples, and simple positive samples have a limited effect on parameter convergence. Thus, to improve the model's effect, we had to make the model focus more on hard positive samples. Therefore, in this paper, we propose using Focal loss, which was introduced by Lin et al. [30], instead of the original loss function.

The cross-entropy loss function for ordinary binary classification is calculated as follows:

$$CE(p, y) = \begin{cases} -\log(p) & y = 1 \\ -\log(1 - p) & otherwise \end{cases} \tag{5}$$

where $p$ is the predicted probability of the sample in the class, $y$ is the sample label, and $y = 1$ means $y$ is a positive sample. If Equation (8) is used to represent $p$, standard cross-entropy can be simplified and expressed as Equation (9).

$$P_t = \begin{cases} p & y = 1 \\ 1 - p & otherwise \end{cases} \tag{6}$$

$$CE(p, y) = CE(p_t) = -\log(p_t) \tag{7}$$

However, the weights in standard cross-entropy are the same for both positive and negative samples, so they will be affected by an imbalance of positive and negative samples,

resulting in the model being influenced by a large number of negative samples with poor results. To balance the positive and negative samples, the balanced cross-entropy loss function used by Yao et al. [31] introduces a balancing factor, $\alpha_t$, and $\alpha_t$ is calculated as in Equation (8), where $\alpha \in [0, 1]$. The balanced cross-entropy is calculated as shown in Equation (9). The EAST model uses the balanced cross-entropy loss function. The specific calculation formula is shown in Equation (10) [16]:

$$\alpha_t = \begin{cases} \alpha & y = 1 \\ 1 - \alpha & otherwise \end{cases} \tag{8}$$

$$CE(p_t) = -\alpha_t \log(p_t) \tag{9}$$

$$\begin{aligned} L_s &= balanced - xent(\hat{Y}, Y^*) \\ &= -\beta Y^* \log \hat{Y} - (1 - \beta)(1 - Y^*) \log(1 - \hat{Y}) \end{aligned} \tag{10}$$

where $\hat{Y} = F_s$ represents the predicted score map, and $Y^*$ represents the true value. $\beta$ is the balancing factor between positive and negative samples, and the formula is calculated as follows:

$$\beta = 1 - \frac{\sum_{y^* \in Y^*} y^*}{|Y^*|} \tag{11}$$

Although the balanced cross-entropy loss function improves the imbalance between positive and negative samples, it still does not solve the imbalance between hard and easy samples. Focal loss, in order to adjust both positive and negative samples and hard and easy samples, adds a modulating factor to balanced cross-entropy, and its equation [25] is as follows:

$$FL(p_t) = -\alpha_t(1 - p_t)^\gamma \log(p_t) \tag{12}$$

When a sample is a simple sample, $p_t$, close to 1, the modulating factor $(1 - p_t)^\gamma$ is close to 0, decreasing the loss contribution of the simple sample. Conversely, when the bounding box is misclassified, $p_t$ is relatively smaller, the modulating factor is close to 1, and its contribution to the loss calculation is nearly unaffected by the modulating factor. This way, the model can focus more on the hard positive samples and enhance the model's efficacy. The parameters in the Focal loss calculation formula are set to $\alpha_t = 0.25$ and $\gamma = 2$ according to the literature [25].

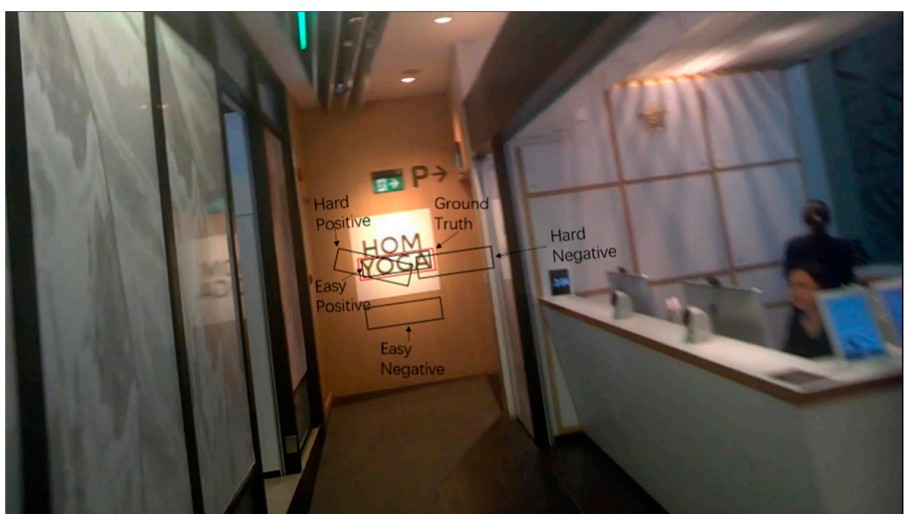

**Figure 7.** Four sample classification examples.

## 4. Experiment

### 4.1. Experimental Environment and Evaluation Metrics

The experiments in this study were developed on a Window 10 operating system using PyCharm, and the deep learning framework was pytorch1.0.1. The hardware environment was: Intel Corei9 (2.80 GHz) processor, 64 GB memory, and NVIDIA RTX 2080 SUPER (8 G memory) graphics card.

The evaluation metrics were precision, recall, and F1 score, which were calculated as follows. TP represents the number of samples that were positive and predicted to be positive; FP represents the number of samples that were actually negative and predicted to be positive; FN represents the number of samples that were actually positive and predicted to be negative, and TN represents the number of samples that were actually negative and predicted to be negative.

$$
\begin{aligned}
prescision &= \frac{TP}{TP+FP} \\
recall &= \frac{TP}{TP+FN} \\
F1 &= 2 * \frac{precision*recall}{precision+recall}
\end{aligned}
\tag{13}
$$

### 4.2. Experimental Steps

The experiment in this study had two main steps: constructing the dataset and training and validating the model. The overall process is shown in Figure 8.

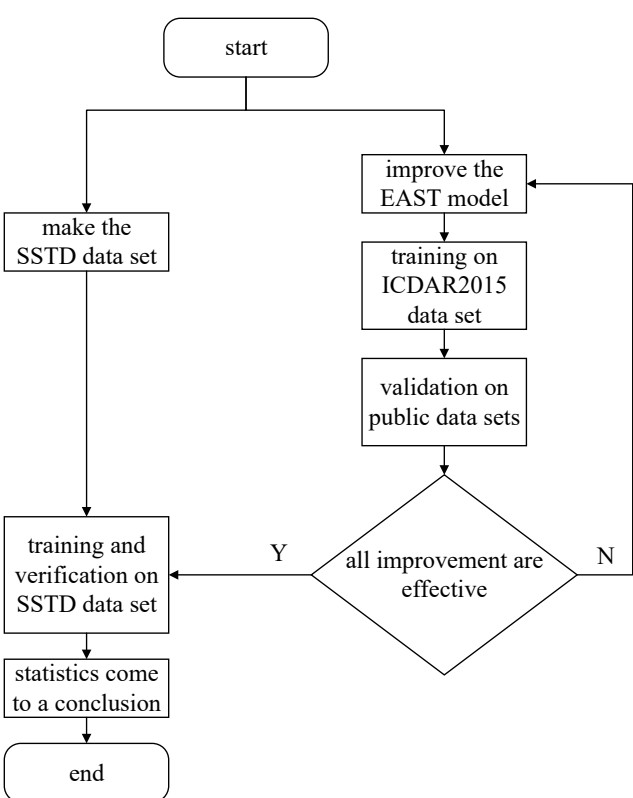

**Figure 8.** Experimental flow chart.

### 4.3. Construction of SSTD Dataset

The SSTD dataset constructed in this study contained a total of 1000 pictures taken on actual roads and publicly available street sign-related pictures on the Internet, including pictures of street signs on both sides and above roads, highway road signs, scenic road signs, and other scenes. For a street sign scene image, which is greatly influenced by the brightness of ambient light and the noise of daily shooting, the collected pictures were expanded by randomly increasing or decreasing the brightness and randomly increasing Gaussian noise to the dataset. A total of 2000 pictures were obtained, 1500 of which were

used for the training set and 500 for the test set. Select pictures from the SSTD dataset are shown in Figure 9.

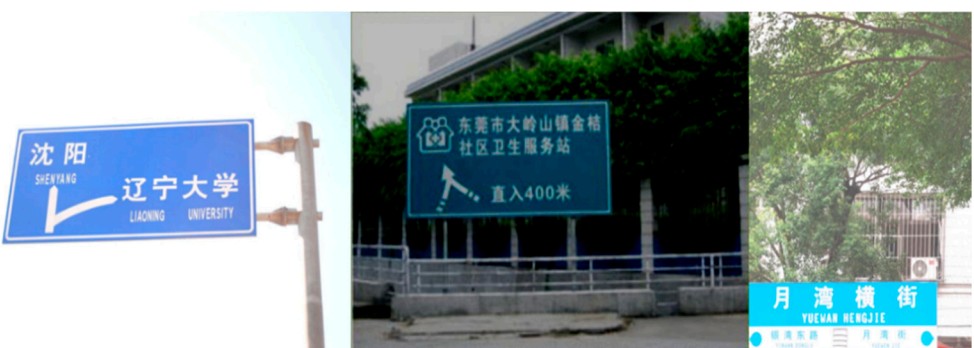

**Figure 9.** Select pictures from the SSTD dataset.

### 4.4. Experiment and Analysis

The public dataset used in this study was the ICDAR2015 dataset, which contained 1500 images taken in natural environments containing text. There were 1000 images in the training set and 500 images in the test set. The input image size was uniformly scaled to the VGG16 network model trained on ImageNet, which was used as the pre-training model. Moreover, the batch size was set to 6, the initial learning rate was set to 0.001, and the learning rate was reduced by a multiplier of 0.1 for every 300 epochs, with a minimum learning rate of 0.00001, until the model performance no longer improved.

This study's ablation experiments were trained and tested on the ICDAR2015 dataset. The effects of the three improvements on the EAST model proposed in this paper are shown in Table 1. The EAST model with VGG16 as the core network is used as the reference object, and the rest of the improved model's core network also uses VGG16. As can be seen from Table 2, changing the loss function to Focal loss rather than balanced cross-entropy improved the precision, recall, and F1 scores by enhancing the model's learning for hard, positive samples. By improving the shrinkage strategy of label generation, the model more accurately predicted the short edge of the bounding box; thus, the precision and F1 scores improved. After adding FEM, although the precision decreased to some extent, the recall was improved from 73.52% to 76.94% because of the increased ability to detect long text, and the final F1 score was improved from 79.95% to 81.03%, thereby improving the overall model performance. As seen from Table 1, the effects of this paper's proposed improvements to the EAST detector were verified. As shown in Table 3, the results of this paper's improved EAST model were compared with other models on the ICDAR2015 dataset. In terms of recall, the proposed model was inferior to the original EAST model using the deeper ResNet50 as the core network. However, owing to the proposed model's advantages in precision, the F1 score and overall performance were nevertheless better than those of the other models.

**Table 2.** Comparison of the effects of ablation experiments.

| Model | Precision (%) | Recall (%) | F1 (%) |
|---|---|---|---|
| EAST + VGG16 | 80.50 | 72.80 | 76.40 |
| EAST + Focal loss | 82.28 | 73.56 | 77.68 |
| EAST + Focal loss + improved shrinking algorithm | 87.61 | 73.52 | 79.95 |
| EAST + Focal loss + improved shrinking algorithm + FEM | 85.59 | 76.94 | 81.03 |

**Table 3.** Comparing detection effects of the improved EAST model with other models on the ICDAR2015 dataset.

| Model | Precision (%) | Recall (%) | F1 (%) |
|---|---|---|---|
| CTPN + VGG16 | 74.20 | 51.60 | 60.90 |
| Seglink + VGG16 | 73.10 | 76.80 | 75.00 |
| WordSup | 77.03 | 79.33 | 78.16 |
| Yao et al. [31] | 72.26 | 58.69 | 64.77 |
| EAST + VGG16 | 80.50 | 72.80 | 76.40 |
| EAST + ResNet50 | 77.32 | 81.66 | 79.43 |
| EAST + PAVNET2x | 83.60 | 73.50 | 78.20 |
| EAST + PAVNET2x MS | 84.64 | 77.23 | 80.77 |
| STN-OCR [32] | 78.53 | 65.20 | 71.86 |
| Poly-FRCNN-3 [33] | 80.00 | 66.00 | 73.00 |
| RFPN-4s [34] | 85.10 | 76.80 | 80.80 |
| Ours | 85.59 | 76.94 | 81.03 |

Figure 10 compares the detection effects of the EAST model with this paper's improved model on the ICDAR2015 dataset. It can be seen that the improved EAST model was more accurate in generating bounding boxes of long text compared with the original model.

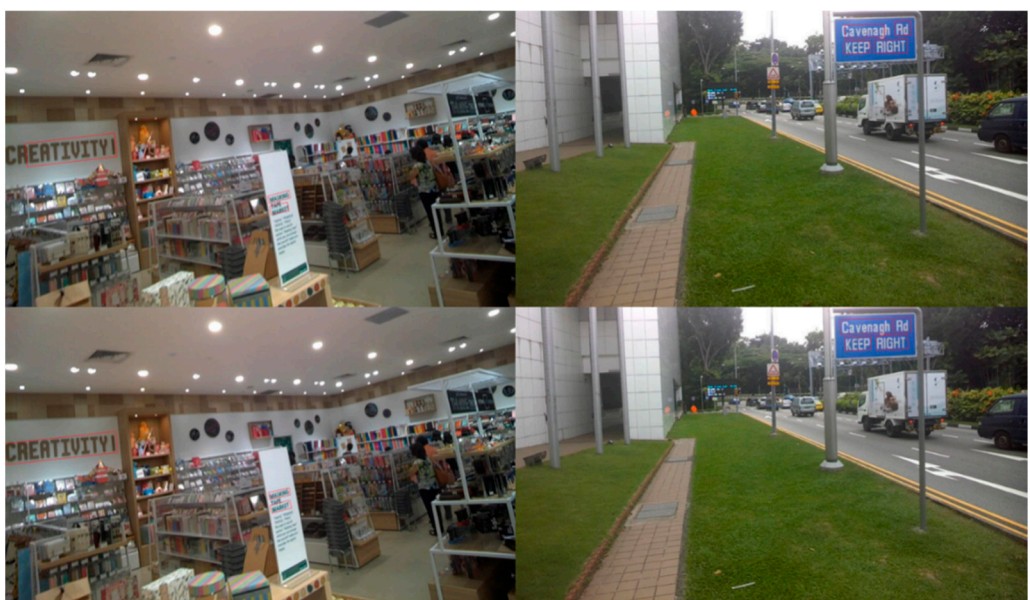

**Figure 10.** Comparison of detection effect between the EAST model and improved EAST model on the ICDAR2015 dataset.

Table 4 and Figure 11 compare the detection effects of the original EAST model with this paper's improved model on the SSTD dataset. On the SSTD dataset, the improved EAST model saw higher scores in both precision and recall compared with the EAST model, increasing the F1 score from 78.84% to 81.48% and overall performance. The comparison in Figure 11 shows that the improved EAST model was indeed more accurate in predicting longer text regions compared to the EAST model, which verified the effectiveness of this paper's proposed EAST model improvements. At the same time, the performance of the model on the SSTD data set verifies that the model proposed in this paper can solve the three challenges proposed in the second section. Our model can detect images obtained from multiple shooting angles in a bright or dark environment. Images of different resolutions contain multiple types of text such as Chinese characters, English, numbers, and punctuation marks.

**Table 4.** Comparing detection effects of the improved EAST model with the EAST model on the SSTD dataset.

| Model | Precision (%) | Recall (%) | F1 (%) |
| --- | --- | --- | --- |
| EAST + VGG16 | 73.80 | 84.61 | 78.84 |
| Ours | 78.30 | 84.94 | 81.48 |

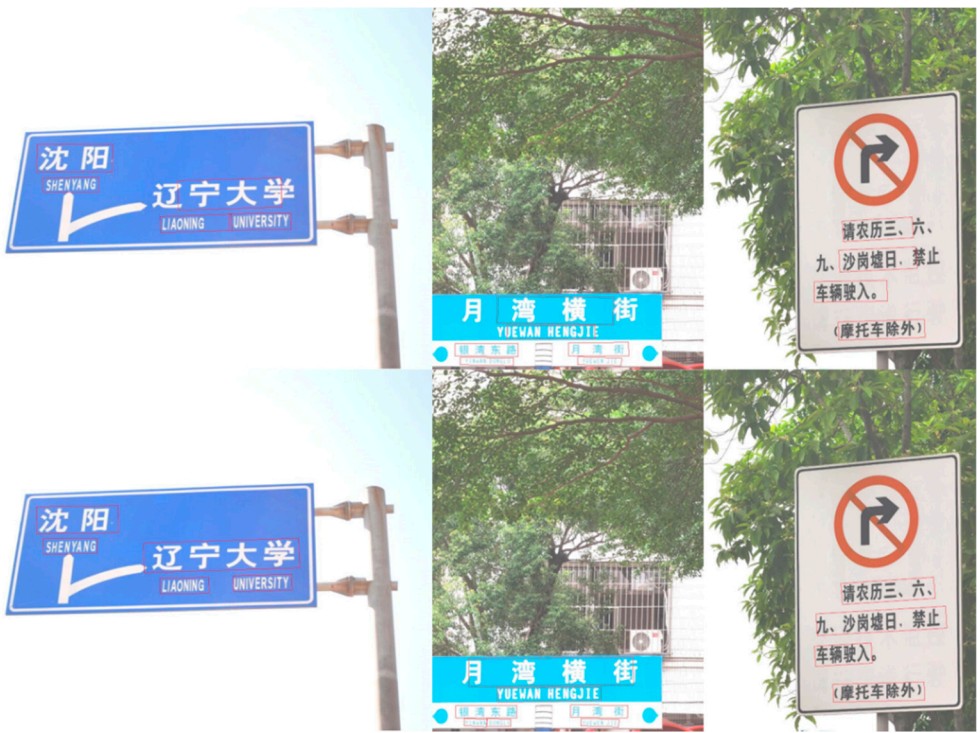

**Figure 11.** Comparison of detection effect between the EAST model and the improved EAST model on the SSTD dataset.

*4.5. Discussion*

There has been a trend toward using deeper feature extraction networks such as ResNet50, and adding complex structures such as bidirectional feature pyramids, to improve text detection model performance. In this paper, grounded in the purpose of practical application, we do not introduce complex structures to improve model performance in order to avoid an increasing number of parameters and computational burden. Instead, our FEM significantly increased the receptive field while only introducing very small increases in parameters. After calculation, the parameter of the original EAST model is 15.09 MB, and the parameter of each FEM module is 0.62 MB, that is, the four FEM modules introduce a total of parameters. It is 2.48 MB. It can be seen that the improvement of this article does not introduce too much parameter. The improved EAST model proposed in this paper inherits the structural advantages of the EAST model, and our model doesn't require pre-designed anchor frames. From the experimental results in Tables 2–4, this paper's modified EAST model improved upon the EAST model without using deeper feature extraction networks and increasing structural complexity. Overall performance of the improved EAST model was higher compared with other models, which verified that this model is an efficient street sign text detection model.

## 5. Conclusions and Future Prospects

This paper produced the SSTD dataset for street sign natural scenes and, based on analysis of existing natural scene text detection models and the characteristics of street sign text scenes, proposed an improved EAST model based on the original EAST detector.

To address the EAST model's deficiencies in detecting long text regions because of small receptive fields, three improvements were proposed: altering the shrinkage calculation during label generation, adding a feature enhancement module (FEM), and changing the loss function approach to Focal loss. Compared with the original EAST model, the improved model increased the F1 score from 76.40 to 81.03 on the ICDAR2015 dataset, and from 78.84 to 81.48 on the SSTD dataset. The improved EAST model's detection effects for long text regions were also enhanced. Ultimately, this paper proposed an improved EAST model with the better overall performance for text region detection in street sign scenes.

In future research, based on the work we did before, on the one hand, we will expand the data set with pictures containing Chinese characters of minority nationalities, so that the model can detect more kinds of characters. On the other hand, we will tackle text recognition from regions with high tilt angles and perspective distortion, as well as street sign scenes that contain Chinese characters, English letters, numbers, and punctuation marks. Then we will propose a model that recognizes the text detected by the improved EAST model, to propose a combined model that can fully detect and recognize street sign text from end to end. In addition, we will conduct research on the application of our work in the field of assisted navigation for autonomous driving and intelligent translation.

**Author Contributions:** The authors' contributions are summarized below. M.L., Y.M. and C.-L.C. made substantial contributions to the conception and design. M.L. and Y.M. were involved in drafting the manuscript. M.L., Y.M. and Q.T. acquired data and analysis and conducted the interpretation of the data. The critically important intellectual content of this manuscript was revised by C.-L.C. All authors have read and agreed to the published version of the manuscript. The authors would like to thank the anonymous reviewers and the editors for all the helpful suggestions.

**Funding:** This research was funded by the National Social Science Fund of China, grant number (20BGL141).

**Institutional Review Board Statement:** Not applicable.

**Informed Consent Statement:** This study is only based on theoretical basic research. It does not involve human subjects.

**Data Availability Statement:** The data used to support the findings of this study are available from the corresponding author upon request.

**Conflicts of Interest:** The authors declare no conflict of interest.

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
