# Peer review of "An Efficient Text Detection Model for Street Signs"

_applsci, doi:10.3390/app11135962_

Round 1

Reviewer 1 Report

In our present world, it is extremely important to have systems that are capable of recognizing written texts and interpreting them for the user. An improved version of the EAST detection procedure in the publication could indeed be a significant step in this direction. Modification procedures to avoid the detection problems in this article were only partially able to achieve the desired results. Future developments drawn in this regard, however, may provide an opportunity to provide solutions to further problems that may arise.

Author Response

In our present world, it is extremely important to have systems that are capable of recognizing written texts and interpreting them for the user. An improved version of the EAST detection procedure in the publication could indeed be a significant step in this direction. Modification procedures to avoid the detection problems in this article were only partially able to achieve the desired results. Future developments drawn in this regard, however, may provide an opportunity to provide solutions to further problems that may arise.

Authors’ Response:

Thank you very much for the reviewer’s comments on our work. In Section 5, we propose that we will expand the data set to support more language character detection, as well as the recognition of the detection results obtained by the model proposed in this paper. Eventually, we will study how to apply our work to areas such as autonomous driving assistance and intelligent translation. What we have revised has been shown in lines 464-466 and lines 471-472 of the revised draft.

Reviewer 2 Report

The authors proposed a method for detecting texts for street signs. Based on the well-known text detection model, EAST, they made some modifications and proposed improved EAST. Although the proposed model is a minor modification of an existing method, experimental results show its superiority compared to some existing methods.

The authors claimed that one of the problems of the original model is ineffective in detecting long text regions and proposed to improve the bounding box's shrinking algorithm. This kind of strategy is introduced by other text detection algorithms. For example, the authors in the following paper proposed to use anchors suitable for texts and multiple region-proposal-networks to solve the same kind of problem. The authors should survey papers more related to the proposed method and should discuss the advantage of the proposed method.
Nagaoka, et al., "Text Detection Using Multi-Stage Region Proposal Network Sensitive to Text Scale," Sensors, vol.21, 2021.

In the experiment section, the authors only compared their method with old ones. They should compare their method with more recent ones. In addition, the authors mentioned the number of parameters and computational cost. I understand that they made efforts in reducing the number of parameters. However, quantitative evaluation in terms of the number of parameters and computational cost should be conducted.

Page 2, lines 20-28, section numbers are mistaken.

In some references, the first and last name of the authors are mistaken.

Author Response

Reviewer 2 Comments:

The authors proposed a method for detecting texts for street signs. Based on the well-known text detection model, EAST, they made some modifications and proposed improved EAST. Although the proposed model is a minor modification of an existing method, experimental results show its superiority compared to some existing methods.

Authors’ Response:

Thank you for your recognition of our work.

The authors claimed that one of the problems of the original model is ineffective in detecting long text regions and proposed to improve the bounding box's shrinking algorithm. This kind of strategy is introduced by other text detection algorithms. For example, the authors in the following paper proposed to use anchors suitable for texts and multiple region-proposal-networks to solve the same kind of problem. The authors should survey papers more related to the proposed method and should discuss the advantage of the proposed method.
Nagaoka, et al., "Text Detection Using Multi-Stage Region Proposal Network Sensitive to Text Scale," Sensors, vol.21, 2021.

Authors’ Response:

Thank you very much for the reviewer's comments. In section 2 and table 1, we add analysis of related papers in lines 129-132 and line 135.

In the experiment section, the authors only compared their method with old ones. They should compare their method with more recent ones. In addition, the authors mentioned the number of parameters and computational cost. I understand that they made efforts in reducing the number of parameters. However, quantitative evaluation in terms of the number of parameters and computational cost should be conducted.

Authors’ Response:

In the experiment section in table 3, we add data on the effects of some new models in line 411. And in section 4.5, we add the discussion on the amount of module parameters in lines 442-445 and lines 446-447.

Page 2, lines 20-28, section numbers are mistaken.

Authors’ Response:

The chapter number error on the second page has been corrected in lines 66-73.

In some references, the first and last name of the authors are mistaken.

Authors’ Response:

In the references, errors in the author's first name and last name have been corrected in lines 488-555.

Reviewer 3 Report

Authors propose an "efficient text detection model for street signs ". Although the topic is of great interest, I do have the following comments:

1- The English of the paper should be improved.

2- Authors use words that are not used in scientific writing such "Good detection for long text areas". What does "good" mean? how "good" is being measured?

3- The authors claim that "the goal of this study was to propose a text region detection model with the simplest possible structure, smaller parameter numbers, less computation, and better performance for the detection and recognition of text in street sign scenes." Did they really prove all of these claims? 

4- In page 4, authors list 3 issues with the current methods and claim that their method overcomes theses methods. 

5- Claims in points 3 and 4 above should be addressed (in the Experiment and analysis and Discussion sections) one  by one to make the claims valid.  Having some of the claims discussed does not make the other claims valid!

Author Response

Reviewer 3 Comments:

1- The English of the paper should be improved.

Authors’ Response:

Thank you very much for the reviewer's comments, we've asked LetPub to polish the article. We added the edited Certificate for your reference.

2- Authors use words that are not used in scientific writing such "Good detection for long text areas". What does "good" mean? how "good" is being measured?

Authors’ Response:

It can be measured by the accuracy and recall rate of model detection, that is, the model can detect long text areas, and the prediction box position is accurate. This statement can be verified by Tables 2, 3 and 4.

3- The authors claim that "the goal of this study was to propose a text region detection model with the simplest possible structure, smaller parameter numbers, less computation, and better performance for the detection and recognition of text in street sign scenes." Did they really prove all of these claims? 

Authors’ Response:

In section 4.5, it shows that the simple structure of our model is because our model inherits the advantages of the simple structure of the EAST model, that is, there is no need to design the anchor frame; because the amount of calculation is not easy to quantify, the calculation amount is removed, and the parameter amount is added in section 4.5 Note: The data in Tables 3 and 4 can verify that the model proposed in this paper has better performance than the original model, and is at a better level than other models.

4- In page 4, authors list 3 issues with the current methods and claim that their method overcomes theses methods. 

5- Claims in points 3 and 4 above should be addressed (in the Experiment and analysis and Discussion sections) one by one to make the claims valid.  Having some of the claims discussed does not make the other claims valid!

Authors’ Response:

The pictures in the SSTD data set proposed in this article have the problems raised on the fourth page, and the performance of the model on this data set confirms that our model can better deal with these challenges. This statement is added to lines 419-424.

Round 2

Reviewer 2 Report

The authors have addressed the reviewer's comments and the manuscript has been improved.

Reviewer 3 Report

Authors have addressed all my comments and the paper is publishable now. Thank You!